# High Prevalence of Gestational Diabetes Mellitus in Rural Tanzania—Diagnosis Mainly Based on Fasting Blood Glucose from Oral Glucose Tolerance Test

**DOI:** 10.3390/ijerph17093109

**Published:** 2020-04-29

**Authors:** Louise Groth Grunnet, Line Hjort, Daniel Thomas Minja, Omari Abdul Msemo, Sofie Lykke Møller, Rashmi B. Prasad, Leif Groop, John Lusingu, Birgitte Bruun Nielsen, Christentze Schmiegelow, Ib Christian Bygbjerg, Dirk Lund Christensen

**Affiliations:** 1Diabetes and Bone-Metabolic Research Unit, Department of Endocrinology, Rigshospitalet, 2100 Copenhagen, Denmark; line.hjort@regionh.dk; 2Section of Global Health, Department of Public Health, University of Copenhagen, 1422 Copenhagen, Denmark; sofiem@sund.ku.dk (S.L.M.); iby@sund.ku.dk (I.C.B.); dirklc@sund.ku.dk (D.L.C.); 3Center for Pregnant Women with Diabetes, Department of Obstetrics, Rigshopsitalet, 2100 Copenhagen, Denmark; 4National Institute for Medical Research, Tanga Center, Tanga 5004, Tanzania; minjartd@gmail.com (D.T.M.); mtengeti@gmail.com (O.A.M.); jpalusingu@gmail.com (J.L.); 5Department of Clinical Sciences, Clinical Research Centre, Lund University, 22100 Malmø, Sweden; rashmi.prasad@med.lu.se (R.B.P.); leif.groop@med.lu.se (L.G.); 6Finnish Institute of Molecular Medicine (FIMM), Helsinki University, 00014 Helsinki, Finland; 7Department of Obstetrics and Gynaecology, Aarhus University Hospital, N 8200 Aarhus, Denmark; birgitte.bruun.nielsen@regionh.dk; 8Centre for Medical Parasitology, Department of Immunology and Microbiology, University of Copenhagen, N 2200 Copenhagen, Denmark; chsch@sund.ku.dk; 9Department of Gynaecology and Obstetrics, Juliane Marie Centre, University Hospital Rigshospitalet, 2100 Copenhagen, Denmark

**Keywords:** prevalence, gestational diabetes, Tanzania, haemoglobin concentration

## Abstract

Gestational diabetes mellitus (GDM) is associated with poor pregnancy outcomes and increased long-term risk of metabolic diseases for both mother and child. In Tanzania, GDM prevalence increased from 0% in 1991 to 19.5% in 2016. Anaemia has been proposed to precipitate the pathogenesis of GDM. We aimed to examine the prevalence of GDM in a rural area of Tanzania with a high prevalence of anaemia and to examine a potential association between haemoglobin concentration and blood glucose during pregnancy. The participants were included in a population-based preconception, pregnancy and birth cohort study. In total, 538 women were followed during pregnancy and scheduled for an oral glucose tolerance test (OGTT) at week 32–34 of gestation. Gestational diabetes mellitus was diagnosed according to the WHO 2013 guidelines. Out of 392 women screened, 39% (95% CI: 34.2–44.1) had GDM, the majority of whom (94.1%) were diagnosed based solely on the fasting blood sample from the OGTT. No associations were observed between haemoglobin or ferritin and glucose measurements during pregnancy. A very high prevalence of GDM was found in rural Tanzania. In view of the laborious, costly and inconvenient OGTT, alternative methods such as fasting blood glucose should be considered when screening for GDM in low- and middle-income countries.

## 1. Introduction

The prevalence of gestational diabetes mellitus (GDM) is increasing worldwide [1]. Since GDM is associated with a higher risk of adverse pregnancy outcomes as well as with long term adverse consequences for both mother and child including an increased risk of developing metabolic diseases such as type 2 diabetes [2,3], it is important to identify these high-risk women. Today, few studies have examined the prevalence of GDM in Sub-Saharan Africa (SSA) and diagnosis remains sub-optimal due to associated logistical and cost barriers for resource-constrained populations [4]. A systematic review from 2015 reported prevalence rates up to 14% of GDM but with high heterogeneity between studies [5]. A recently published systematic review and meta-analysis concluded that the pooled prevalence of GDM in Africa was 13.6%, with the highest prevalence in Central Africa (20.4%) and the lowest in Northern Africa (7.6%), estimated by using the current International Association of the Diabetes and Pregnancy Study Groups (IADPSG) diagnostic criteria [6]. In Tanzania, GDM prevalence ranged from 0% in a rural areas in 1991 when applying WHO 1985 criteria [7], to 19.5% in urban areas in 2016 using the WHO 2013 criteria [8]. Furthermore, a study from 2012 showed that the prevalence seems to be dependent on living conditions, with a GDM prevalence of 8.4% in urban areas and 1.0% in rural areas in Tanzania using WHO 1999 criteria [9]. Diagnostic criteria have varied and comparing reports of GDM prevalence across studies can be challenging, but the prevalence seems to have increased over time. Several factors such as pre-pregnancy body mass index (BMI), family history of diabetes, age and glycosuria are all associated with higher risk of developing GDM, but anaemia has also been proposed as playing a role in the GDM pathogenesis [10]. Even though the worldwide prevalence of anaemia has slightly decreased in the last decade, the prevalence in 2011 was above 50% in Central and West Africa and 36% in East Africa among pregnant women [11]; the corresponding figures in Tanzania were 40%–59% [12]. Iron deficiency is the most common nutritional cause of anaemia. Anaemia is usually assessed by haemoglobin (Hb) concentration and iron deficiency by ferritin concentration [13].

In the present study, we aimed to examine the prevalence of GDM in a rural area of Tanzania using a standard oral glucose tolerance test (OGTT), and to examine whether there is an association between Hb and ferritin concentrations in pregnancy and risk of developing GDM.

## 2. Subjects

The participants were included in a large population-based study named FOETALforNCD (Foetal exposure and Epidemiological Transition: the role of Anaemia in early Life for Non-Communicable Diseases in later life) [14]. All participants were residents in the rural Korogwe and Handeni districts in Tanga region, Northeast Tanzania. Mobile clinics were set up in 48 villages in the two districts and seventeen dispensaries functioned as outreach satellite sites. The study population were enrolled from July 2014 to March 2016 and consisted of two different groups. One group (n = 383) was enrolled before pregnancy and conceived during the study period. For the pre-pregnancy cohort the inclusion criteria were; an age between 18–40 years; a negative urine pregnancy test; no usage of modern contraceptive methods except condoms; not having a child less than nine months old; living in an accessible area; and upon conception, willingness to attend antenatal care and give birth at Korogwe district hospital. Women having tried to conceive unsuccessfully for more than two consecutive years (regarded as infertile) were excluded. When conceiving they were enrolled in the pregnancy part of the study irrespective of their Hb levels. The other group consisted of 155 pregnant women, who were recruited ≤14 gestational weeks based on the degree of anaemia; with severe anaemia: Hb ≤ 8 g/dL [4.96 mmol/L], mild-moderate anaemia: Hb 8.1–10.9 g/dL [5.02–6.81 mmol/L] or without anaemia: Hb ≥ 11 g/dL [6.82 mmol/L], Figure 1.

## 3. Materials and Methods

The study has been described in details elsewhere [14] but in short, at inclusion, information on ethnicity and socioeconomic status was obtained. Gestational age was estimated by crown rump length in the first trimester and head circumference in the second trimester using transabdominal ultrasound (5–2 MHz, Sonosite TITAN^®^ and Sonosite Turbo^®^, Sonosite, Bothell, WA, USA) [15,16]. During pregnancy, all women were scheduled for at least four antenatal visits which were equally distributed during pregnancy, at week 20–22, week 26–28, week 32–34 and week 37–39. At all antenatal visits, the women were monitored for anaemia (Hb < 11 g/dL), infections including malaria, hypertensive disorders and anthropometric measurements were also collected. Weight was measured while barefooted and wearing light clothes on a digital weighing scale. Height in centimetres was measured with a stadiometer and BMI was defined as weight in kilograms divided by the square of the height in meters (kg/m^2^). Mid-upper arm circumference (MUAC) was measured on the upper right arm at the midpoint of the acromion process and the tip of the olecranon. Blood pressure was measured by a digital blood pressure monitor (r-champion^®^ N, Rudolf Riester, Jungingen, Germany) with an inflatable cuff of an appropriate size after resting for at least 5 min [17]. Furthermore, the women were screened with a urine dipstick for glucosuria as well as for polyhydramnios (defined as an amniotic fluid index > 20) with transabdominal ultrasound.

In total, 538 women were included in the present study, of whom 392 women were screened with an OGTT, with the majority being screened during weeks 32–34 of gestation. The OGTT was scheduled for the antenatal visit in week 32–34, due to concerns about missing women with GDM if they were screened at the previous visit in week 26–28. Most of the women had their visits in the first available week, i.e., week 26 and week 32. Out of the 538 women, 77 had a miscarriage, 34 moved outside the study area or withdrew their consents and the remaining 35 women either did not comply with the overnight fasting criterion before the examination, missed their visit in the last trimester or gave birth before week 32 (Figure 1). No differences in age, BMI, MUAC or Hb was observed between the women who underwent an OGTT and the ones that were not tested (data not shown).

The OGTT was performed after 12-h of overnight fasting. Venous fasting blood samples were taken and blood from the butterfly blood sampling tube was used to measure glucose immediately using HemoCue^®^ Glucose 201 RT (Ängelholm, Sweden). The coefficient of variation for the HemoCue Glucose 201 has previously been shown to be 1.8%, and is almost as precise as the hexokinase laboratory method [18].

Hereafter, each woman was instructed to drink a glucose solution containing 82.5 g monohydrate glucose (corresponding to 75 g anhydrous glucose) (Rapra’s Pure Glucose, Instant Energy Plus, Rapra Limited, Nairobi, Kenya) in 250 mL of water within 5 min. Thereafter, she was instructed to be seated in the waiting area without drinking, eating or smoking for the following 2 h, where blood glucose was measured by a fingerprick test after 1- and 2-h. If the OGTT was abnormal, the women were given dietary advice and were controlled every third week for fasting blood glucose and glycated haemoglobin (HbA1c) (Afinion AS 100 analyzer). If the women by mistake had not been fasting at their follow-up visit, random blood glucose was measured. Furthermore, ultrasound for foetal growth was performed bi-weekly. A few of the women had more than one OGTT because there was a suspicion of GDM early in pregnancy due to, for example, glucosuria and it was then repeated routinely at weeks 32–34.

Gestational diabetes mellitus was diagnosed according to the WHO 2013 guidelines as follows: fasting venous plasma glucose 5.1–6.9 mmol/L, or venous plasma 1-h glucose value ≥10 mmol/L or 2-h glucose value 8.5–11.0 mmol/L. In the present study, the fasting sample was taken from venous blood whereas 1-h and 2-h samples were based on capillary blood. Using HemoCue^®^ Glucose 201 RT (Ängelholm, Sweden), plasma-equivalent glucose levels were measured. A conversion factor for whole blood-to-plasma glucose of 1.11 was used as recommended by the Internatinal Federation of Clinical Chemistry IFCC [19]. The measurement based on capillary blood was converted using the model by Colagiuri et al. and accordingly, the cut-off value for GDM was 10.7 mmol/L for 1-h values and 9.15 mmol/L for 2-h values [20].

Haemoglobin levels were measured using Sysmex^®^ KX-21N haematological analyser (Sysmex Corporation, Kobe, Japan) on venous blood. Ferritin, albumin and bilirubin (Vista_1500 system) and C-reactive protein (CRP) (Afinion AS 100 analyser) were measured. Further details on the FOETALforNCD study have previously been published in [14].

## 4. Statistics and Ethics

Normally distributed data are presented as mean ±(SD) whereas skewed data are presented as median and interquartile range. Differences between women diagnosed with GDM and non-GDM controls were analysed using Student´s t-test for normally distributed data, Mann–Whitney U test for non-normally distributed data or X^2^ test for categorical data. Pearson correlation analyses were performed in order to explore the association between Hb and ferritin concentrations and fasting glucose measurements at the time when the OGTT was carried out. Furthermore, correlation analyses between Hb at inclusion (before week 11) and fasting glucose from the OGTT was carried out. *p* ≤ 0.05 was considered statistically significant. All statistical analyses were performed using SAS 9.4 statistical software (SAS Institute, Cary, NC, USA).

Ethical clearance was granted by the Medical Research Coordinating Committee of the National Institute for Medical Research (reference number NIMR/HQ/R.8a/Vol. IX/1717). After giving oral information, written informed consent in Swahili (or thumbprints from illiterate women, given after a witness who was not involved in the project was informed) was obtained prior to enrolment. All procedures were conducted in accordance with the Declaration of Helsinki.

## 5. Results

Characteristics of women with GDM and non-GDM controls are presented in Table 1. Women with GDM tended to be older. No difference in gestational age at delivery or birth weight was observed between the two groups. Body mass index at inclusion (before week 11) was in the normal range < 25 kg/m^2^ and no differences in family history of diabetes were found between the two groups (Table 1). Very few had malaria at the time the OGTT was performed (10 GDM women and 12 controls). Seven women, all in the GDM group, had type 2 diabetes diagnosed based on the OGTT during pregnancy (fasting glucose ≥ 7 mmol/L or 2-h OGTT ≥ 11.1 mmol/L).

In total, 392 women completed an OGTT as described. Eighteen women had the OGTT performed twice and one woman underwent an OGTT three times. If GDM was diagnosed in more than one OGTT, blood glucose measurements from the first OGTT were included in the analyses. By the OGTT screening, we found 153 women with GDM, i.e., a prevalence of 39.0% (95% CI: 34.2–44.1). The majority (94.1%) were diagnosed based on the fasting blood sample (Table 2). While 64% of the women had their OGTT performed in week 32+0 to 34+0, 18% had the OGTT before week 32+0 and 18% after week 34+0. If only including women, who had their OGTT performed in week 31–34 (GA 217–238 days) the GDM prevalence was 41% (124/300). Furthermore, no difference between GDM and controls in gestational age at the time when the OGTT was preformed was observed (GDM: 32+5 (2+2) vs. control 33+4 (2+2) weeks+days, *p* = 0.07). In the present study WHO diagnostic cut-points based on the Hyperglycaemia and Adverse Pregnancy Outcomes (HAPO) study were used. It has been demonstrated that the protocol used in the HAPO study results in a mean plasma glucose drop of approximately 0.2 mmol/L compared to samples immediately measured or using glycolytic inhibitors that immediately stabilize glucose [21]. If we take this positive bias of 0.2 mmol into account and raise the cut-off for GDM to 5.3 mmol/L in the present study, a GDM prevalence of 27.6% is found.

At the time of the OGTT, no differences in Hb, ferritin, CRP or albumin levels were seen between the GDM and control groups (Table 1). In addition, no correlations between Hb and fasting glucose (r = −0.08, *p* = 0.13) or between ferritin and fasting glucose (r = 0.09, *p* = 0.11) were found in the combined group including both GDM and control women. When the women were analysed separately according to GDM status, no correlations were seen between Hb and fasting glucose (GDM: r = −0.05, *p* = 0.56 and non-GDM: r = −0.11, *p* = 0.11) or between ferritin levels and fasting glucose (GDM; r = 0.04, *p* = 0.68 and non-GDM; r = 0.12, *p* = 0.13). There was a weak association between Hb at inclusion and fasting glucose when the OGTT was performed (r = −0.09, *p* = 0.09). After adjustment for age and BMI, the association became significant (β −0.06, *p* = 0.04).

## 6. Discussion

In rural Tanzania, we found a GDM prevalence of 39.0%. This prevalence is higher than what has previously been found in SSA, including Tanzania. A systematic review including 22 studies from SSA found a GDM prevalence of up to 14% [5]. However, all the reviewed studies included only six of the total 47 countries from SSA and half of them were from Nigeria [5]. A recent study with data collected during 2015–2016 in Moshi Town, Tanzania, reported a GDM prevalence of 19.5% using the WHO/IADPSG 2013 criteria [8] which was similar to the criteria used in the present study. Some of the discrepancies could be explained by differences in BMI and other body anthropometric indices between the study populations or due to a poorer and unhealthier diet including a lower intake of vegetables and more rice and bread among women from Korogwe and Handeni districts, which is rural and a poorer area as compared with Moshi Town. It can also be speculated that a higher degree of insulin resistance is present in our study population. Alanine aminotransferase (ALAT) is elevated in most cases of non-alcohol fatty liver diseases and ALAT levels have been associated with decreased insulin sensitivity in subjects with type 2 diabetes [23]. On the other hand, bilirubin may protect against insulin resistance by reducing visceral fat [24]. Nevertheless, no differences in either ALAT or bilirubin between the GDM women and control women were observed in the present study.

In the present study, 94.1% of all women were classified with GDM based on a fasting blood sample. This finding is consistent with a large Danish study, where the majority of GDM cases were also classified based on the fasting blood glucose value [25]. Furthermore, it is consistent with a large population-based screening for GDM in North India, where more than 5000 women were screened and more than 94% of GDM cases were identified based on the fasting blood glucose value [26]. Interestingly, when frequencies of GDM were reported among the 15 centres that participated in the Hyperglycaemia and Adverse Pregnancy Outcome (HAPO) study, a substantial centre-to-centre variation was found in which the time-point of glucose measures met the diagnostic threshold for GDM [27]. In places such as Bellflower and Providence in the USA and Barbados, more than 70% of women were classified as GDM by fasting blood glucose, whereas the GDM diagnosis was based on the 1-h value in 64% of the cases in Bangkok, and the 2-h value in 29% of the cases in Hong Kong. This variation between centres has been suggested to be influenced by the degree of obesity and degree of abnormal glucose tolerance in the general populations where the HAPO centres were located. Since most of the women with GDM in the present study had their diagnosis based on the fasting values rather than the OGTT, the applicability of an OGTT for the diagnosis of GDM can be questioned in some low- and middle-income countries given its higher cost and being more time consuming [4]. Therefore, the one size fits all approach in relation to GDM diagnosis is debatable [25]. Accordingly, in our setting in Korogwe, it might be recommendable to perform an accurate measure of fasting blood sample and then only perform an OGTT in those without GDM diagnosed by a fasting blood glucose value.

An ongoing debate is whether universal or selective screening for GDM is the most appropriate approach. In a population from SSA it has been shown that approximately 20% of GDM cases defined by WHO 2013 criteria would have been missed if a selective screening approach was used [28]. Risk-factor based screening including pre-pregnancy BMI > 27 kg/m^2^, family history of diabetes, previous GDM, previous baby >4500 g or glucosuria, followed by an OGTT is the current procedure in Denmark [29]. When the WHO 2013 criteria for diagnosing GDM by fasting values were applied to Danish women without GDM, about 40% turned out to have GDM but had a low risk of pregnancy complications [25]. In the present Tanzanian population, there was no difference in family history of diabetes between GDM and control women and 75% of the GDM women had BMI < 27 kg/m^2^; thus, it seems that when possible, the thresholds for diagnosing GDM and the screening approach should be adopted to the local setting.

In the present study, no associations (except for a weak association between Hb at inclusion and fasting glucose from the OGTT after adjustment for age and BMI) between Hb or ferritin levels and glucose concentrations were found. Previous studies have suggested that anaemia in pregnancy increases the risk of GDM [9,30] whereas a recent meta-analysis based on six studies, reported that pregnant women with iron-deficiency anaemia were 39% less likely to develop GDM [31]. These discrepancies may be due to differences in study populations, diagnosis of GDM and timing of Hb and ferritin measurements. Although the underlying mechanisms linking anaemia and GDM are still unknown, various mechanisms have been suggested such as iron being a catalyst of several biochemical reactions leading to production of reactive oxygen species which can decrease insulin sensitivity [32]. Others suggest that iron overload can be toxic to β-cells and thereby influence insulin secretion [33]. It has been speculated that the association between anaemia and increased risk for GDM may be explained by a more general nutritional status including micronutrient deficiency among women with GDM [31]. However, most women in our study had normal levels of folic acid and vitamin B12.

A limitation in the present study is that the majority of the OGTTs were performed during weeks 32–34, whereas most other studies have been testing for GDM during week 24–28. Since insulin resistance increases during pregnancy, this later testing could contribute to our findings of a higher GDM prevalence. However, during pregnancy, fasting glucose most likely decreases due to an increase in plasma volume in early pregnancy and increased glucose utilization in later pregnancy by the foeto-placental unit [34]. In line with this, there was a tendency towards a higher gestational age at the time of OGTT among women without GDM compared to women with GDM. The majority of GDM cases in the present study were diagnosed based on their fasting glucose value and we believe that this late testing did not significantly influence our results. Even though the study participants in the present study were thoroughly instructed on how to fast there is always a risk that instructions were not followed. However, the fasting blood glucose values were similar, and the 1-h and 2-h glucose values were actually lower in the present study compared to a recent study from Ethiopia where they found a GDM prevalence of 12.8%. This supports the notion that our study participants most likely had been fasting. Finally, since the prevalence of diabetes in the pre-pregnancy cohort was previously shown to be 0.8% [14] and the prevalence of type 2 diabetes in adults in Tanzania is 3.7% according to the International Diabetes Federation (IDF) [35], it is not likely that the high prevalence of GDM observed was due to a large proportion of undiagnosed type 2 diabetes.

Finally, the use of ferritin as a marker for iron deficiency can be challenging if not taking the inflammatory state into account, since inflammation increases ferritin levels [36]. Thus, if iron deficiency is associated with GDM, this might be masked in women with inflammation and “falsely” increased ferritin levels. However, in the present study no difference in CRP was found between the groups.

In the present study, blood glucose was measured by HemoCue 201, where whole blood is converted to plasma glucose concentrations. We are aware that the ratio between whole blood and plasma depends on haematocrit and hydration status and therefore also that anaemia can be a confounder for the glucose levels. However, since no association was found between Hb and glucose levels, we do believe that adjustment for anaemia would not have significantly influenced GDM prevalence in the present study. Furthermore, we are aware that even a small divergence in the way of collecting samples such as using glycolytic inhibitors or on-point measurements, time of fasting, measurement devices etc., cause marginal differences in blood glucose levels, and that small shifts in fasting plasma glucose can translate into a large variation in GDM incidence. In the present study, the sample collection and treatment were all conducted by trained project employees, using standard operating procedures to ensure uniformity throughout the study, and blood samples were analysed immediately which should reduce the bias. However, after having taken a possible positive bias into account, the GDM prevalence was 27.6%, indicating the importance of where to set the cut-points for diagnosis and that it may require revision if different pre-analytical protocols are used. Furthermore, the use of point-of-care testing versus laboratory testing are also highly relevant to consider when examining and comparing GDM prevalence. Malley et al. showed that the diagnostic accuracy of a point-of-care measurement versus a laboratory measurement was 83% (74.2%–89.8%) [37], suggesting that if the tests are performed in resource-rich settings, then the use of blood collection tubes containing citrate that inhibits glycolysis followed by laboratory tests are preferable. However, in low-resource settings such as Tanzania where this procedure is not always possible, it will be relevant to validate point-of-care testing with laboratory testing if new screening procedures are going to be implemented.

Another limitation is that the study was designed to look at anaemia in early life and its impact on foetal and placental development and risk for non-communicable disease later in life, not to examine GDM prevalence per se. The present study was powered to detect an effect of −0.31 z-score on birth weight in 2nd trimester severe anaemia (Hb ≤ 8 g/dL) with a significance level of 0.05, a power of 0.80, and an assumption of 15% lost to follow-up based upon previous observations in the STOPPAM study [38]. Furthermore, we cannot exclude the possibility that the study is underpowered to detect a potential association between Hb and ferritin and fasting glucose.

## 7. Conclusions

In conclusion, prevalence of GDM was high in rural North-Eastern Tanzania. We recommend an introduction of routine screening for hyperglycemia in Tanzania. In view of the laborious, costly and patient-unfriendly OGTT, alternative methods such as fasting blood glucose should be considered when screening for GDM in low- and middle-income countries.

## Figures and Tables

**Figure 1 ijerph-17-03109-f001:**
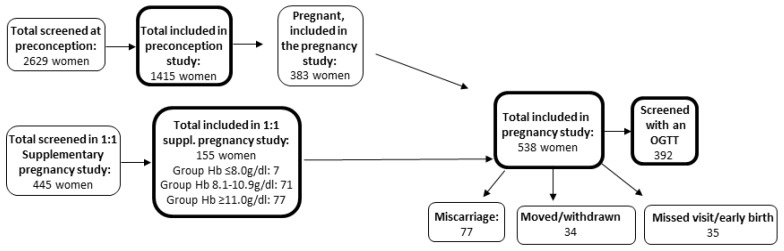
Overview of the recruitment of the study cohort. OGTT: Oral glucose tolerance test.

**Table 1 ijerph-17-03109-t001:** Characteristics of the women who underwent an OGTT.

	GDM Cases	Controls	*p*-Value
	*n* = 153	*n* = 239	
Age (years) *	27 (22;35)	26 (22;33)	0.07
Parity (n: nulliparity/1 child/≥2 children) ^#^	15/34/104	38/46/155	0.21
Ethnicity (n (%)) ^#^			
Sambaa	61 (39.9)	83 (34.7)	
Zigua	47 (30.7)	81 (33.9)	
Pare	11 (7.2)	15 (6.3)	
Bondei	2 (1.3)	8 (3.3)	
Other	32 (20.9)	52 (21.8)	0.65
Education (n (%)) ^#^			
None	11 (7.2)	24 (10.0)	
Incomplete primary	23 (15.0)	32 (13.4)	
Complete primary	103 (67.3)	161 (67.4)	
Secondary or higher	16 (10.5)	22 (9.2)	0.76
Source of domestic water (n (%)) ^#^			
Tap	95 (62.1)	129 (54.0)	
Well	22 (14.4)	37 (15.5)	
River	33 (21.5)	65 (27.2)	
Pond/pool	3 (2.0)	8 (3.3)	0.42
Type of toilet facility (n (%)) ^#^			
Flush	46 (30.3)	61 (25.5)	
Pit	105 (69.1)	175 (73.2)	
None	1 (0.6)	3 (1.3)	0.55
Family history of diabetes	12 (8.7)	19 (9.0)	0.93
BMI at inclusion (kg/m^2^) *	23.4 (20.9;26.9)	22.7 (20.5;25.6)	0.14
MUAC at inclusion (cm) *	27.5 (25.9;30.8)	27.5 (25.1;30.2)	0.27
Fasting glucose (OGTT) (mmol/L)	5.5 (0.6)	4.4 (0.4)	-
1-h glucose (OGTT) (mmol/L)	7.5 (1.4)	6.6 (1.1)	-
2-h glucose (OGTT) (mmol/L)	6.8 (1.1)	6.3 (0.9)	-
Systolic BP (mmHg)	103.9 (9.97)	104.4 (11.75)	0.64
Diastolic BP (mmHg)	66.1 (8.0)	66.3 (9.3)	0.84
Pulse (beats per min.)	92.2 (12.1)	90.8 (11.2)	0.25
Hb (g/dL)	10.63 (1.32)	10.72 (1.27)	0.49
Albumin (g/dL)	28.0 (2.15)	27.72 (2.37)	0.29
Ferritin (ng/mL) *	9.3 (6.0;17.5)	8.5 (6.2;12)	0.19
CRP (mg/L) *	5.0 (5.0;5.0)	5.0 (5.0;5.0)	0.65
ALAT (U/L) *	13.9 (11.4;17.4)	14.0 (11.8;16.9)	0.90
Vitamin B12 (pmol/L) *	277 (216;370)	274 (205;340)	0.28
Folic acid (nmol/L) *	24.7 (16.6;35.6)	25.4 (16.1;38.4)	0.61
Bilirubin (µmol/L) *	6.0 (4.6;9.1)	5.7 (4.3;8.4)	0.28
Malaria	10 (6.5)	12 (5.1)	0.65
HIV seropositive	5 (3.3)	4 (1.7)	0.14
Gestational age at birth (weeks+days)	39+6 (1+4)	40+0 (1+4)	0.26
Birth weight (g)	3012 (539)	3014 (455)	0.97

Values are mean (SD), median (25th;75th percentiles) or *n* (%). *p*-values are calculated using student’s T-test, Mann–Whitney U test * or X^2^ test ^#^. The measurements have been obtained at time of OGTT if anything else is not mentioned. Abbreviations: GDM: gestational diabetes, OGTT: oral glucose tolerance test, MUAC: mid upper arm circumference, CRP: C-reactive protein, Hb: haemoglobin, BP: Blood pressure.

**Table 2 ijerph-17-03109-t002:** Percent of GDM diagnosed by each glucose measure.

GDM Diagnosis Based upon:	n	% (95%-CI)
**Fasting sample**	144	94.1 (94.0;94.2)
**2-h OGTT**	5	3.3 (3.2;3.4)
**Fasting sample and 1-h OGTT**	3	2.0 (1.9;2.0)
**Fasting sample and 1-h OGTT and 2-h OGTT**	1	0.65 (0.61;0.70)

95%-CIs of the proportion are calculated using Wilson score method [22].

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
