# Peer review of "High Prevalence of Gestational Diabetes Mellitus in Rural Tanzania—Diagnosis Mainly Based on Fasting Blood Glucose from Oral Glucose Tolerance Test"

_ijerph, 2020, doi:10.3390/ijerph17093109_

Round 1
Reviewer 1 Report
High Prevalence of Gestational Diabetes Mellitus Rural Tanzania – Diagnosis Mainly Based on Fasting Blood Glucose from Oral Glucose Tolerance Test International Journal of Environmental Research and Public Health
GDM is one of main complications during pregnancy, which also is key issue in maternal and child health care. The present study focused on the prevalence of GDM in rural Tanzania and some interesting findings were showed, especially showing the importance of fast blood glucose testing in screening. However, some technical issue about design and data analysis should be addressed further.
- The higher prevalence of GDM was found in rural Tanzania, 39% was higher. Authors should discuss possible reasons.
- In the methods, authors should present the estimation of sample size for estimating the prevalence of GDM. Some rationality about sample size should be mentioned.
- A technical issue should be pointed out that in this birth cohort 538 pregnant women were included but actually 392 women were screened for GDM. It meant that there were almost 30% of women missing for testing. Therefore, authors should present the difference in background information between women included and women excluded from testing. If some big difference will be found, conclusion should be treated cautiously.
- Authors found the higher percentage of women were screened out for GDM based on fasting blood glucose, so they should discuss more about it, why? Present discussion just provided some facts.
- Authors found no/weak associations between Hb or ferritin levels and glucose concentrations were found. But they seemed not to consider other potential confounders. Of course, it may be result from different study population or smaller sample size. It is suggested that more analysis about this issue could be involved.
Reviewer 2 Report
Thank you for submitting your article “High Prevalence of Gestational Diabetes Mellitus in Rural Tanzania – Diagnosis Mainly Based on Fasting Blood Glucose from Oral Glucose Tolerance Test” for review. Overall this is a straight forward and well written article. Please find below my constructive commentary which I hope can improve your work prior to publication.
Abbreviation- note please ensure that all abbreviations are written with capitals: e.g. Gestational diabetes mellitus (GDM) should be Gestational Diabetes Mellitus (GDM), same applies to OGTT, BMI etc.
Be consistent. Line 1 of introduction. gestational diabetes (GDM) = Gestational Diabetes Mellitus (GDM)
The prevalence of gestational diabetes (GDM) is increasing worldwide ( insert reference)
A recently published systematic review and meta-analysis concluded that the pooled prevalence of GDM in Africa was 13.6%, with the highest prevalence in Central Africa (which was what? insert here) and lowest in Northern Africa (which was what? please insert here).
“Women having tried to conceive unsuccessfully for more than two consecutive years (regarded as sub-fertile) were excluded” is this sub-fertile a clinical label and definition? Please cite reference.
Im not sure- but it just reads awkward labelling women as sub-fertile, and so I wonder if you can justify this definition somehow?
on the degree of anaemia; with severe anaemia: Hb≤8g/dl [4.96 mmol/L], mild-moderate anaemia: Hb 8.1-10.9 g/dL [5.02-6.81 mmol/L] or without anaemia: Hb ≥11g/dL [6.82 mmol/L], Figure 1
Degree of anaemia- cite source for categories here please
((5-2 MHz,Sonosite TITAN® and Sonosite Turbo®, Sonosite, Bothell, WA, USA))
Remove double brackets
Once you stated an abbreviation please use this consistently, e.g. GDM
Discussion point- ??? of the 538 women, 77 had a miscarriage – is this ~14%a verage/expected for the region?
Overall well written and concise, very minor changes recommended prior to publication.
Round 2
Reviewer 1 Report
Authors have addressed most of my comments, and mansucript has been improved much. However, sample size estimation or power analysis should be re-considered. Authors said in the introduction part that "In the present study, we aimed to examine the prevalence of GDM in a rural area of Tanzania using a standard oral glucose tolerance test (OGTT), and to examine whether there is an association between Hb and ferritin concentrations in pregnancy and risk of developing GDM." It means present study wanted to know the HB between and GDM. So power analysis should focus on it. Maybe the original study aimed to test the impact of anaemia on birth weight, but they should clearify rationality that the present sample is enough to test the association between Hb and ferritin concentrations in pregnancy and risk of developing GDM. Therefore, I suggest that power analysis of present sample size should be presented, and some discussion should be included on it.
Author Response
Please see attachement
